# Aerosol Distributions and Sahara Dust Transport in Southern Morocco, From Ground-Based and Satellite Observations

Hassan Bencherif [1,*], Aziza Bounhir [2,3], Nelson Bègue [1], Tristan Millet [1], Zouhair Benkhaldoun [2], Kévin Lamy [1], Thierry Portafaix [1] and Fouad Gadouali [4]

1 Laboratoire de l'Atmosphère et des Cyclones—LACy, UMR 8105, CNRS, Université de La Réunion, Météo-France, 97490 Saint-Denis, France; nelson.begue@univ-reunion.fr (N.B.); tristan.millet@univ-reunion.fr (T.M.); kevin.lamy@univ-reunion.fr (K.L.); thierry.portafaix@univ-reunion.fr (T.P.)

2 High Energy Physics and Astrophysics Laboratory, Cadi Ayyad University, Marrakech 40000, Morocco; a.bounhir@uca.ma (A.B.); zouhair@uca.ma (Z.B.)

3 Faculté des Sciences, Université Mohamed V, Rabat 10014, Morocco

4 Direction Générale de la Météorologie, Beni Mellal 23020, Morocco; gadoualif@gmail.com

* Correspondence: hassan.bencherif@univ-reunion.fr

**Abstract:** The present study investigates aerosols distributions and a strong Sahara dust-storm event that occurred by early August 2018, in the South of Morocco. We used columnar aerosol optical depth (AOD), Angstrom Exponent (AE) and volume size distributions (VSD) as derived from ground-based observations by 2 AERONET (AErosol RObotic NETwork) sun-photometers at Saada (31.63°N, 8.16°W) and Ouarzazate (30.93°N, 6.91°W) sites, over the periods 2004–2019 and 2012–2015, respectively. The monthly seasonal distributions of AOD, AE, and VSD showed a seasonal trend dominated by the annual cycle, with a maximum aerosol load during summer (July–August) and a minimum in winter (December–January), characterized by a coarse mode near the radius of 2.59 μm and a fine mode at the radius of 0.16 μm, respectively. Indeed, this study showed that aerosol populations in southern Morocco are dominated by Saharan desert dust, especially during the summer season. The latter can sometimes be subject of dust-storm events. The case study presented in this paper reports on one of these events, which happened in early August 2018. The HYSPLIT (HYbrid Single Particle Lagrangian Integrated Trajectory) model was used to simulate air-mass back-trajectories during the event. In agreement with ground-based (AERONET sun-photometers) and satellite (CALIOP, MODIS and AIRS) observations, HYSPLIT back-trajectories showed that the dust air-mass at the 4-km layer, the average height of the dust plume, has crossed southern Morocco over the Saada site, with a westward direction towards the Atlantic Ocean, before it changed northward up to the Portuguese coasts.

**Keywords:** aerosol optical depth; Sahara dust; Morocco; AERONET; dust transport

## 1. Introduction

Atmospheric aerosol micro-physical and optical parameters are still a major source of error in the estimation of climate change models. Aerosols have a confirmed large impact on regional climate and therefore on the global climate [1]. Atmospheric aerosol characterization is a very important research topic to improve climate models, atmospheric studies, and astronomical observations. Due to Morocco's geographical location in north-west Africa and its landscape, its aerosol populations occur in different types and origins, with a dominance of desert aerosols in the south. Aerosol particles, including Sahara mineral dust, travel in all directions over large areas. In fact, due to its location and latitudinal extent, from 21°N to 36°N, Morocco is subject to a variety of weather and climate conditions modulated by desert winds coming from the South-East and by Atlantic and Mediterranean advections [2]. The country climate indeed varies between a semi-arid Mediterranean

climate in the north and an arid climate in the south, passing through a more temperate climate along the coastline, which becomes drier as one moves into the plains [3]. As reported in many studies, desert dust outbreaks contribute directly to air pollution, impacting air quality by increasing particulate matter concentrations, which indeed has many adverse effects on human health, notably increased mortality risk [4–6]. The Sahara is one of the hottest regions in the world, with mean annual temperatures higher than 30 °C [7], extending zonally over 5000 km, from the Atlantic Ocean to the Red Sea, between Mediterranean North Africa and sub-Saharan Africa. It is the largest desert in the world, with a surface area of 9.1 million km² (31% of Africa), crossing a dozen states, including Morocco. Sandstorms usually occur in the Sahara during summer, under very strong winds generated by the abrupt difference in air density between warm and cold air masses [8]. Apart from their sporadic and seasonal occurrences, the generating processes of these storms are not well understood, as atmospheric conditions make observations difficult and investigations are often very limited in desert regions. In these remote desert regions, satellite observations play a major role, allowing to observe from space aerosol dust in the atmosphere and its transport to other regions. Morocco has a large desert territory including much of the Sahara, which is the largest source of mineral dust emissions in the world influencing North African countries, as well as southern Europe and eastern America [9,10]. According to Engelstaedter et al. [11], North Africa is the place of the main sources of dust and its annual emissions ranges from 170 to 1600 Tg per year. The combination of rich sources of erodible materials and energetic wind systems facilitates the dust long-range transport. The Saharan dust is mainly transported westward into the Atlantic Ocean [12–14]. Moreover, dust is also transported northward across the Mediterranean Sea into southern and central Europe and, in some extreme cases, towards the shores of the Baltic Sea [15–22]. In this work, we studied a case study of Sahara dust transport in southern Europe through the Atlantic Ocean. Few studies have been published on aerosols in southern Morocco so far. Among them, the most important Saharan dust observations conducted in this region are the SAMUM (Saharan Mineral Dust Experiment) [23] and CHARMEX (Chemistry and Aerosols Mediterranean Experiment) [24], measurement campaigns carried out in 2007 and 2013, respectively. All previous studies highlighted that the frequency of desert dust is very low (less than two per year) [25], with most of the events occurring during spring and mainly during summer [26]. Generally, due to the Sahara Desert, North Africa accounts for the major source of dust aerosol, with a relative contribution of about 50% of the total worldwide generation [27]. In addition to their effects on cloud properties and precipitation [28], dust aerosols play important roles in biogeochemical cycles [29], atmospheric chemistry [30], visibility, and human health [31]. Various photometric measurement campaigns allowed to determine the desert dust volume size distribution with a radius between 0.05 and 15 µm [32,33], but for a limited period of time, no longer than 2 years of observation. According to Tahiri et al. [33], from 2013 AERONET measurements, the volume size distributions at 3 Moroccan sites (Oujda, Saada, and Ouarzazate) showed the same shape with median radii around 0.15 µm for fine particles and 2.59 µm for coarse particles. However, they found a slight increase in the concentration of large particles in Saada, especially in summer and a low concentration of fine mode aerosol at Ouarzazate. The content of the present paper is organised as follows. After the introduction, in Section 3 we analyse the longest series of ground aerosol observations recorded in southern Morocco at the Saada (2004–2019) and Ouarzazate (2012–2015) sites. These observational data have been very little utilized, or during limited periods of time not exceeding 2 years. The aim in this study is to perform a seasonal analysis of the micro-physical and optical properties of aerosol populations characteristic over the study region, known to be particularly exposed to Saharan dust raised during sandstorms. Section 4 of the paper reports and discusses a case study of a dust event that occurred in early August 2018. In that regard, we combined ground-based observations from sun-photometer measurements in the south of Morocco (at Saada and Ouarzazate) and Portugal (at Cabo da Roca), with satellite observations from MODIS, CALIOP and

AIRS, together with reanalysis data from MERRA-2 and back-trajectory simulations by the HYSPLIT model.

## 2. Observation Data, Method, and Location

### 2.1. Localisation

The Saada site (31.63°N, 8.16°W, 420m asl) is located 10 km and 50 km away from Marrakech and the High Atlas Mountains, respectively. Marrakech is the most important touristic city in Morocco (about 3 million visits in 2019), and a major economic area with nearly 1.5 million inhabitants. For comparison, we used AERONET data measured in Ouarzazate (30.93°N, 6.91°W, 1136m asl). The latter is a small town located 120 km southeast of Marrakech, with almost no industrial activity. The two sites are separated by the High Atlas Mountains, with the highest peak Mont Toubkal at 4170 m. Ouarzazate is thus more exposed to desert winds, with a pre-Saharan climate characterized by low rainfall and a hot and dry summer [34]. Figure 1 shows the location of Morocco in north-west Africa and the geographical locations of the two study sites, Saada and Ouarzazate.

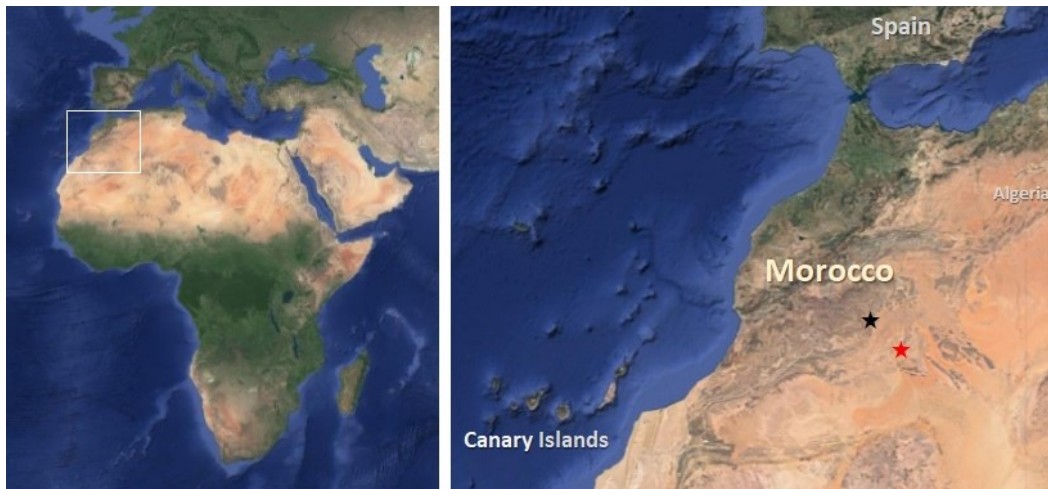

**Figure 1.** Geographical map with the study sites, Saada (31.63°N, 8.16°W) and Ouarzazate (30.93°N, 6.91°W) indicated with black and red stars, respectively. The picture on the left is a zoom on the study area indicated by the white box on the right.

### 2.2. Ground-Based Observations from AERONET

AERONET is an international ground-based network for measuring columnar optical and micro-physical properties of atmospheric aerosols. It is also useful for satellite retrieval validation [35,36]. The network consists of more than 800 observation stations worldwide. The data collected are widely used for monitoring aerosol variations and their optical properties, as well as for assessing the associated radiative forcing and its impact on climate at both regional and global scales. Retrieval of Aerosol Optical Depth (AOD) is based on the measurement of the attenuation by the atmosphere of incoming solar radiation. A clear atmosphere without aerosols is characterized by a maximum of global irradiance, a minimum of diffuse irradiance and a very low AOD near to zero. The Cimel Electronic radiometer (CE318) has been used worldwide for more than two decades in order to provide highly accurate ground-based measurements of the AOD. It performs direct sun and sky measurements, both within several programmed sequences. The direct sun measurements are made in eight spectral bands (340, 380, 440, 500, 670, 870, 940, and 1020 nm). The 940-nm channel is used for column water abundance. Optical depth is calculated from spectral extinction of direct beam radiation at each wavelength based on the Beer–Lambert–Bouguer Law. Attenuation due to Rayleigh scatter, absorption by ozone, and gaseous pollutants are estimated and removed to retrieve the AOD. The sampling interval between measurements is typically 15 min. More details about the performance of the Cimel instrument and AOD retrieval process can be found in several published works [35,37–39]. In addition to the

direct solar irradiance measurements, the Cimel radiometer measures the sky radiance in four spectral bands (440, 670, 870, and 1020 nm), which are inverted following the Dubovik and King method [40] to provide aerosol properties of size distribution and phase function. The determination of the atmospheric aerosol volume size distribution is obtained from sky radiance and the inversion of spectral optical depth measurements. The method is based on discretizing the spectrum in 22 points covering the range variation of both fine and coarse modes [40,41]. The present study is based on the analysis of AOD measurements (level 2.0) recorded at Saada and Ouarzazate sites in the southern Morocco, between 2004–2019 and 2012–2015, respectively. The Level 2.0 of AERONET data is a "Quality Secured" label, which means that the AOD values are pre- and post-field calibrated, automatically cloud cleared, and manually inspected. In addition to the AOD, the associated Angstrom Exponent (AE) and Volume Size Distrib ution (VSD) from Dubovik and King method, are used to characterize the observed aerosol type. The AE is an optical parameter that shows how the optical thickness of aerosols depends on the wavelength, and contains size information on all optically active aerosols in the field of view of a sun-photometer. It is inversely related to the average size of the particles: close to zero or negative for large particles, and greater than one for small ones. O'Neill et al. (2001) revealed that the particle size distribution inversions are constrained to produce small errors in almucantar radiance. This constraint translates into $0.09 < r_{eff} < 0.46$ μm, where the parameter $r_{eff}$ is the best estimate of an effective optical radius, and the upper and lower limits correspond to the neighbourhood of the classical extinction efficiency peak at 500 nm. Indeed, the daily AOD, AE, and VSD values were used to derive the corresponding monthly seasonal trends. The obtained values were used for comparison with daily observations recorded during the sandstorm event that took place in early August 2018. In fact, AE and VSD enable to discriminate between fine and coarse particle modes, which helps to determine their type and presumably their probable origin. Given the geographical position of the study sites, the observed aerosol population is expected to show bimodal distributions, more or less dominated, depending on the season, by the coarse mode from Sahara dust. AOD, AE, and VSD time-series recorded at Saada and Ouarzazate sites were downloaded from the AERONET website (https://aeronet.gsfc.nasa.gov (accessed on 5 April 2022). Figure 2 summarises the level 2 aerosol AERONET data used in the present study. The upper graph (Figure 2a superimposes the number of observations per month for both sites Saada and Ouarzazate. Apart from a few periods of interruption, the Saada site offers a relatively regular series spread over  15 years, with a number of observations per month of about 18. As for the observations at Ouarzazate, they are limited in time coverage (2012–2015), but with a higher density of measurements per month, of the order of 20. The time variations of the AOD recorded over Saada at 440 nm ($AOD_{440}$) are shown in Figure 2b (red line). They highlight the preponderance of the annual cycle, with a minimum in winter and a maximum in summer. The figure also highlights four major events with large AOD values, (higher than 1.2): one event in 2010, two events in 2017, and one event in 2018. In this study we investigate the most recent event, which took place in early August 2018. It is indicated in the figure by the grey background. Superimposed on Figure 2b are the $AOD_{440}$ time series obtained for the site during the same period, from MODIS observations (blue) and MERRA-2 reanalyses (green), presented in the next subsection. Both MODIS and MERRA-2 series show similar AOD variations to those obtained from ground-based observations by the Cimel radiometer. We will present in Section 4 the daily AERONET versus MODIS and MERRA-2 comparisons, and the dust storm event of early August 2018.

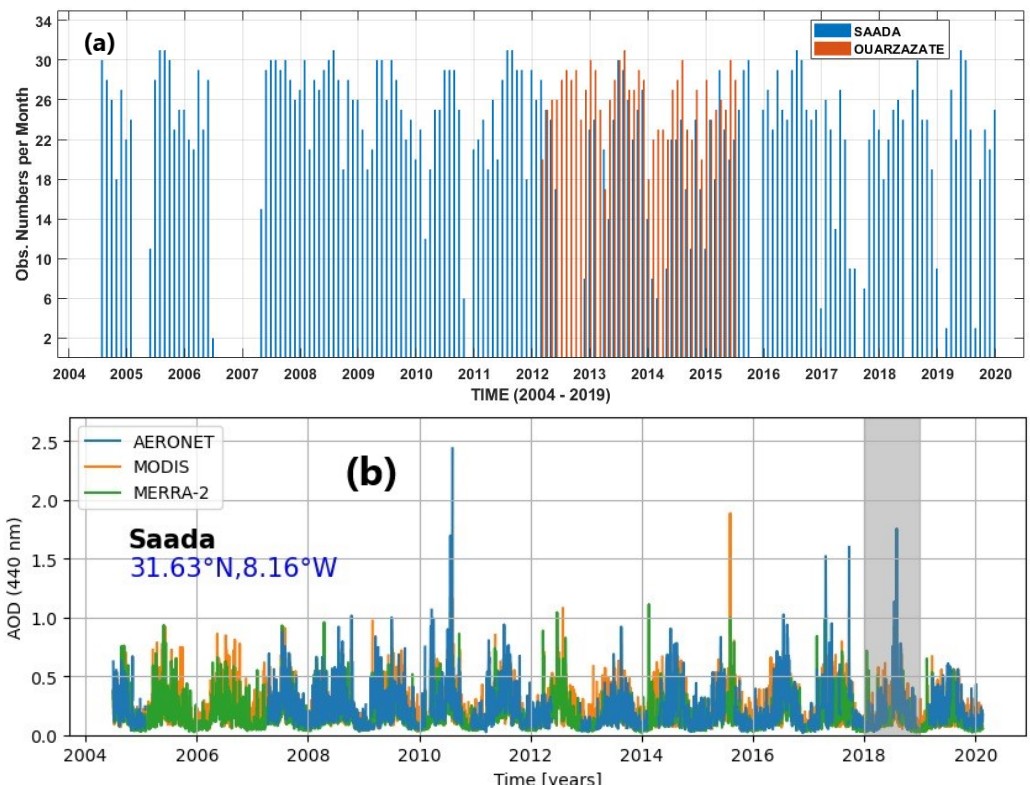

**Figure 2.** (**a**) Number density of aerosol measurements by AERONET sun-photometers at Saada (blue bars) and Ouarzazate (orange bars), from 2004 to 2019. (**b**) Daily $AOD_{440}$ time-series as recorded at Saada from sun-photometer (AERONET level 2.0 data) (red line), from MODIS observations (blue line) and from MERRA-2 assimilations (dashed green line). The grey area frames the year of the study event (2018).

### 2.3. Satellite and Assimilation Datasets

In addition to AERONET ground based records, satellite observations are used to characterize the aerosol loads over the study sites and the surrounding regions, on a seasonal basis and during the dust event. For comparison with ground-based data, AOD from the Moderate Resolution Imaging Spectrometer (MODIS) and from the Modern-Era Retrospective analysis for Research and Applications version 2 (MERRA-2) are used. The MODIS instruments have been operating since 2002 and 2000, respectively, on-board the NASA Aqua and Terra satellites [42,43]. It records data in 36 discrete spectral bands ranging from 0.4 μm (400 nm) to 14.4 μm (14,400 nm) with a spatial resolution that varies between 250 m and 1 km depending on the channel being used. It covers a wide range of different types of atmospheric data such as cloud cover, amount of water vapour in atmospheric columns, vertical distribution of temperature, and aerosol properties. For the purpose of the study, we used MODIS AOD level 3.0 data distributed on the $1° \times 1°$ horizontal grid, downloaded from the Giovanni platform (https://giovanni.gsfc.nasa.gov/giovanni (accessed on 5 April 2022)). The MERRA-2 data is an ensemble of atmospheric re-analyses compiled based on global observations from satellite and ground-based measurements. The MERRA-2 provides meteorological parameters since 1979, including aerosol products [44]. We used the MERRA-2 AOD time series obtained over Saada site at 550 nm. The latter is retrieved by the MERRA-2 algorithm, which is based on a combination of observations and outputs from a forecast model [45]. The MERRA-2 AOD used in this work were also downloaded from the Giovanni platform, and consisted of hourly measurements distributed on a $0.5° \pm 0.625°$ grid. In order to monitor the vertical distribution of aerosol layers and types over the study region, we used the total attenuated backscatter and the associated aerosol subtype products derived from the Cloud-Aerosol Lidar with Orthogonal

Polarization (CALIOP) aboard the CALIPSO satellite [46–48], together with the dust score from the Atmospheric Infrared Sounder (AIRS) aboard the Aqua satellite. By using a dust-detection algorithm, DeSouza Machado et al. [49] showed that the combination of the Atmospheric InfraRed Sounder (AIRS) data from the 4-μm (400 nm) and 10-μm (10,000 nm) channels allowed to retrieve a Dust Score, day and night over ocean and land. Both channels are respectively sensitive to optical depth and to the height of the dust layer. Indeed, the AIRS Dust Score layer indicates the level of atmospheric aerosols. It is computed in form of a numerical scale, which is a qualitative representation of the presence of dust, and an indication of where large dust storms may form. We used the Dust Score to examine dust transport during the dust-storm event that happened in early August 2018. CALIOP products were downloaded from the CALIPSO website (https://www-calipso.larc.nasa.gov (accessed on 5 April 2022)), while the AIRS Dust Score layers are available on the NASA Earth Data portal (https://earthdata.nasa.gov (accessed on 5 April 2022)).

### *2.4. Hysplit Simulations*

In addition to aerosol data and products obtained from ground based and satellite observations, the Hybrid Single Particle Lagrangian Trajectory (HYSPLIT) model was used to simulate air-mass back-trajectories during the dust event over the study region. The HYSPLIT model is commonly used to compute forward and backward trajectories. It uses meteorological data to calculate simple air-mass trajectories, as well as complex simulations of transport, dispersion, chemical transformation, and deposition of gaseous or particulate pollutants [50,51]. Simulations performed in this study are based on the use of a fixed reference frame for the advection of air parcels from their initial positions to create ensemble-back-trajectories [51]. It is assumed that chemistry along the path of the back-trajectory is negligible.

## 3. Results

The results of the seasonal analysis of the AOD and AE observations at Saada and Ouarzazate are superimposed in Figure 3. For both sites the daily data were aggregated and averaged monthly over the 2004–2019 and 2012–2015 periods, respectively. Overall, the monthly AOD values at Saada are higher than those at Ouarzazate, but, as shown in the figure, both sites depict the same seasonal distributions dominated by an annual cycle, showing a minimum AOD in winter (December–January) and a maximum in summer (July–August). The superimposed vertical bars represent the standard deviations associated with each monthly value and indicate the variability of the observed parameter (here AOD and AE). Despite the greater number of observations used for the Saada site, one may note a higher variability of the AOD for this site, in comparison with Ouarzazate, especially for the months of February and August. These observations show that the two sites admit the same AOD seasonal variability, but with different intensities. The obtained difference in terms of intensity is expected due to the fact that the sites are located to the east and west of the High Atlas Mountains (see Figure 1). The latter can modulate the transport of aerosols in both directions, and thus limit the anthropogenic particles transfer eastward (over Ouarzazate) and, in a reverse way, limit the westward travel of the Sahara dust (over Saada). Indeed, the site of Ouarzazate seems cleaner and less exposed to anthropogenic particles, but potentially more subject to desert dust overflow. This is consistent with the AE monthly values obtained for both sites (red curves in Figure 3). Indeed, this figure shows similar seasonal variations in phase opposition with the AOD curves, but lower for the Ouarzazate site, with minimums obtained in summer (July–August) and maximums in winter (December–January). These results support the hypothesis that both sites are under the effect of the coarse mode, presumably desert dust, and that Ouarzazate is more exposed to this mode than the Saada site. When considering the monthly mean VSD distributions for the study sites depicted on Figure 4, one notices the bimodal distributions for both sites, with a quasi-permanent fine mode with a radius of 0.15 μm, and a prominent

occurrence of a coarse mode during the summer months (July–August), with a radius of 2.5 μm. The results in Figure 4 are consistent with those in Figure 3 and underline that the aerosol content over southern Morocco is modulated by desert dust load, especially during summer. There are very few comparative studies of the present work in the literature. The most extensive one is based on the analysis of 2 years of aerosol records on 2012 and 2013 at 3 AERONET sites in Morocco including Saada and Ouarzazate, published by Tahiri et al. [33]. They reported, for both sites, Saada and Ouarzazate, an annual cycle with the minimum and maximum AOD values during winter and summer, characterized by a fine mode at 0.16 μm and a coarse mode near 2.59 μm, respectively. After this seasonal analysis of aerosol distributions, the following section aims to investigate and discuss the case study of a Sahara dust storm that took place in early August 2018 in southern Morocco.

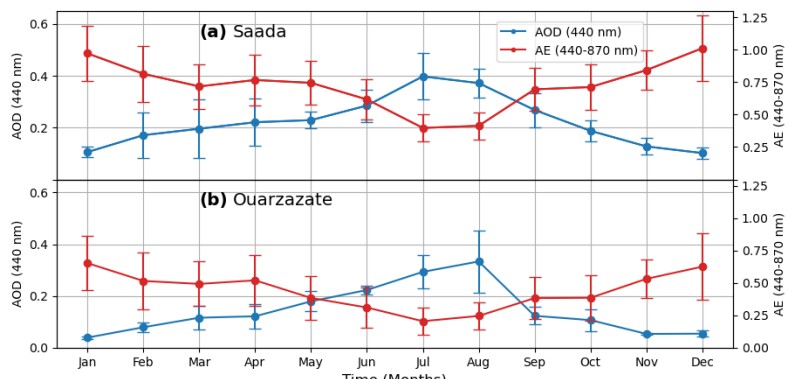

**Figure 3.** Monthly averaged AOD at 440 nm wavelength (blue line with the vertical axis on the left) and AE at the 440–870 nm wavelength range (red line with the vertical axis on the right) values as derived from AERONET data at Saada (**a**) and Ouarzazate (**b**) sites. The vertical bars indicate the corresponding standard-deviations.

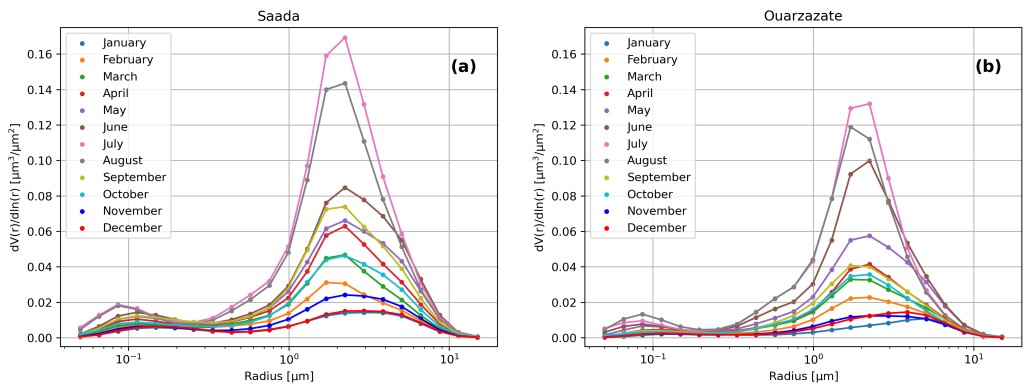

**Figure 4.** Monthly averaged volume size distributions (VSD) obtained from sun-photometer measurement at Saada (**a**) and Ouarzazate (**b**) AERONET sites with level 2.0 data during the study periods (2004–2019 and 2012–2015), respectively.

## 4. Discussion on a Dust-Storm Event in Early August 2018

The aim of this section is to investigate and discuss a case study of the dust-storm event that occurred in southern Morocco early August 2018. The event was initially detected from continuous sun-photometer observations at Saada. As it can be seen in Figure 3, it is the second most intense peak of the entire observation period. It is also the most intense and recent peak of the 2011–2019 decade.

### 4.1. Aeronet Ground-Based Data

As shown in Figures 2 and 3, the AOD in southern Morocco is dominated by an annual cycle, with the maximum occurring in summer (July–August). Figure 5a displays

the daily evolution of the $AOD_{440}$ (red line) measured by sun-photometer at Saada during the period from 5 July to 30 August, which frames the event. It highlights a maximum $AOD_{440}$ (1.77) measured on the 215th day of the year 2018 (2 August). This AOD peak is more than 4 times higher than the average monthly value (0.38) and was also captured by satellite observations from the MODIS experiment and by MERRA-2 reanalyses, with lower amplitudes, 0.9 and 1.0, respectively. Overall, over the observation period, MODIS and MERRA-2 seem to underestimate $AOD_{440}$, especially during the peaks. However, MERRA-2 seems to be in better agreement with the ground-based AERONET observations. These results are consistent with the finding reported by Millet et al. [52] in their comparative study for the same region. To determine the type of aerosols during the event, the daily VSD derived from AERONET observations at Saada between 1 and 5 of August were superimposed on Figure 5b. For comparison, the monthly seasonal VSD profile for August was superimposed (thick red line). The scatter plot in Figure 5c gives the density number plot between AOD (at 440) and AE (at 440–870) values during the period framing the dust event (15 July–15 August). The points obtained for the 1 and 2 August are located at the bottom and on the right side of the plot (low AE and high AOD) which is indicative of the coarse mode. Indeed, plots (a–c) of Figure 5 confirm that the dust-storm event developed between 1 and 3 August and was characterized by a prevailing coarse mode, with a radius in the range 1.5–2.5 μm.

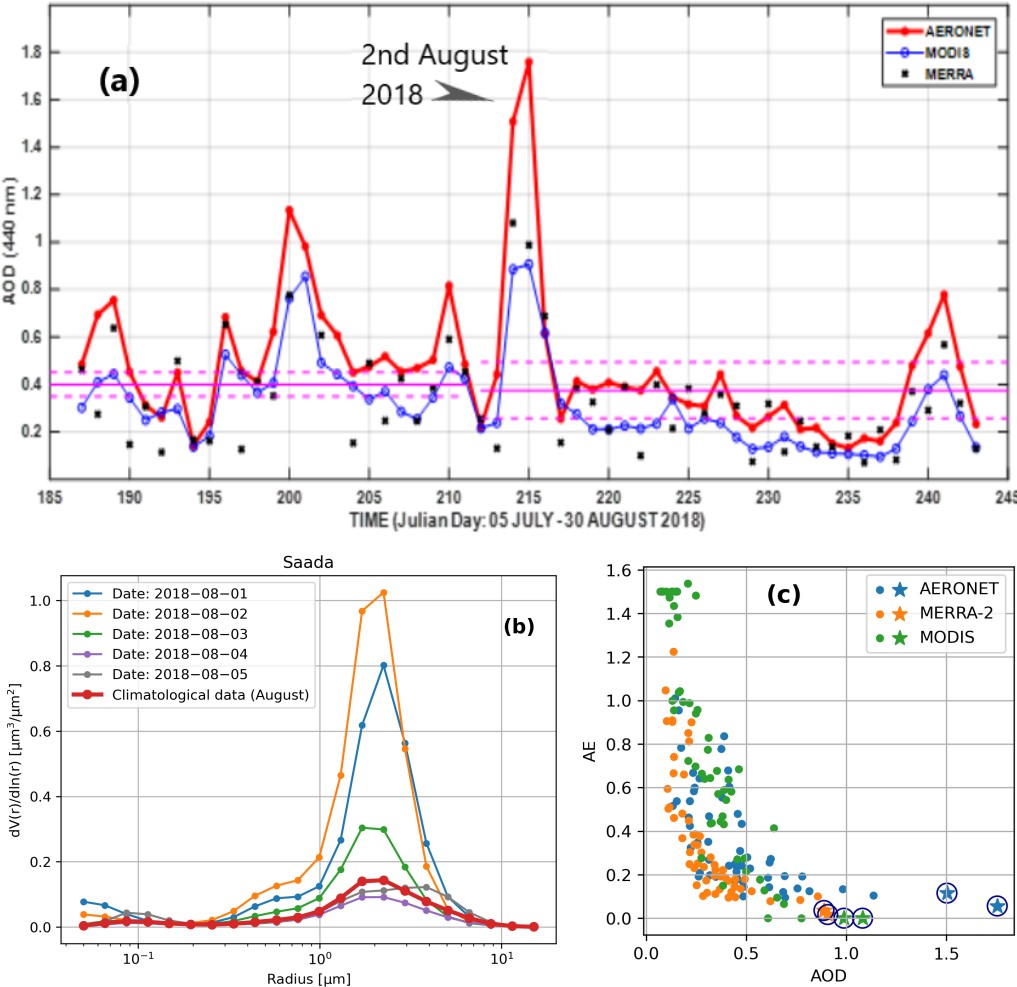

**Figure 5.** (**a**) Daily averaged aerosol optical depth at 440-nm ($AOD_{440}$) from sun-photometer (AERONET) at Saada site, together with MODIS (blue line) and MERRA-2 (black cross symbols) values, from 5 July to 30 August 2018. The corresponding monthly seasonal AOD values from AERONET data are superimposed with horizontal lines framed by the ±1 sigma (standard-deviation)

lines. (**b**) Daily volume particle size distributions obtained from level 2.0 data during the dust event from 1 to 5 August 2018, from the sun-photometer measurement at Saada AERONET site. The monthly seasonal VSD profile for August is superimposed (red heavy line). (**c**) Density number plot between AOD at 440 nm and AE at 440–870 nm during the period from 15 July to 15 August 2018 over Saada from AERONET sun-photometer (blue symbols), MERRA-2 (orange symbols) and MODIS (green symbols) values. The star symbols with a circle indicate the values obtained during the dust event on 1 and 2 August 2018.

*4.2. Satellite Observations and HYSPLIT Simulations*

As it can be seen on the image depicted in Figure 6, the MODIS visible channels captured a large plume of dust from the Sahara Desert travelling westward across south of Morocco and the Atlantic Ocean, on 2 August 2018. From the image, we can observe that the dust plume covered a large part of Morocco and over-flowed westward to the Atlantic coast and over the Canary Islands (Lanzarote and Fuerteventura islands). Pulled by the lower troposphere circulation over the ocean, the plume pathway changed northward and up to the Portuguese coast. In order to investigate the vertical distribution, the aerosol plume was tracked down by the CALIOP instrument on board of CALIPSO satellite, shown by the use of two over-passes over the study area between 2 and 3 August. CALIOP gave the total attenuated backscatter at 532 nm and the associated aerosol subtype by applying the cloud–aerosol discrimination algorithm [53,54]. Figure 7 shows the vertical aerosol distributions during the two tracks (overlaid in the upper-left corner of each panel). From both total attenuated backscatter and aerosol subtype distributions, it comes that the dust plume was located in the lower troposphere in the 3–5 km altitude range, with a north-west trajectory over the Atlantic Ocean. As mentioned in the previous section, we used the Dust Score, which is a qualitative representation of the presence of dust, and indicates where large dust storms may form and the areas that could be affected with. Dust Score distributions shown on Figure 8 are derived from AIRS observations during the dust-storm event, consecutively on 31 July, 1, 2, and 3 August 2018. Apart from the 31 July dust score map, the day before the storm, the following days' maps are consistent with the timing of the event, and the plume overpass over southern Morocco and over the Saada site. Despite their qualitative significance, the Dust Score maps display the dust outflow trajectories, and its north and north-westward transport to the Atlantic and Portuguese coasts. Moreover, the Dust Score distributions of 2 and 3 August indicate that the origin of the storm was further south-east in the Sahara, between southern Algeria and northern Mali. We used the HYSPLIT model in its ensemble-back-trajectory mode initialized on 2 August at a secondary site—Lisbon, Portugal. The ensemble-trajectory option starts multiple trajectories from the Lisbon start site. The resulting trajectories were calculated using the Global Data Assimilation System (GDAS) database for the plume mean height (4 km above ground level) as derived from CALIOP observations (see Figure 7). They are superimposed on Figure 9a. They are consistent with the MODIS, CALIOP, and AIRS observations (Figures 6–8) and indicate that the dust plume in the 4-km layer over Lisbon has crossed southern Morocco over the Saada site. Furthermore, as it appears on Figure 9, the starts of the simulated HYSPLIT trajectories show that the dust plume was located, three days prior, in the south and east of Morocco, and crossed a large area of the Sahara Desert extending from Libya to Morocco. To analyse the variation of AOD over Lisbon during the dust event, we superimposed daily aerosol optical depth at 550-nm (AOD$_{550}$) measured by the AERONET sun-photometer at Cabo da Roca (38.78°N; 9.50°W, 136m asl), 40 km west of Lisbon, together with AOD$_{550}$ values from MODIS and MERRA-2, during the 15 July–15 August 2018 period.

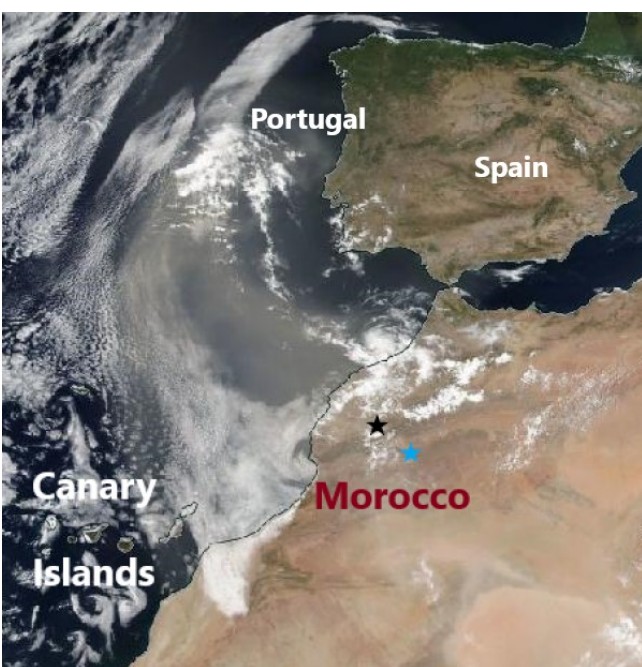

**Figure 6.** True colour image from MODIS/Terra visible channels on 2 August 2018 showing a dust storm travelling over the south of Morocco at 11.50 UTC. Saada and Ouarzazate AERONET sites are indicated with black and blue stars.

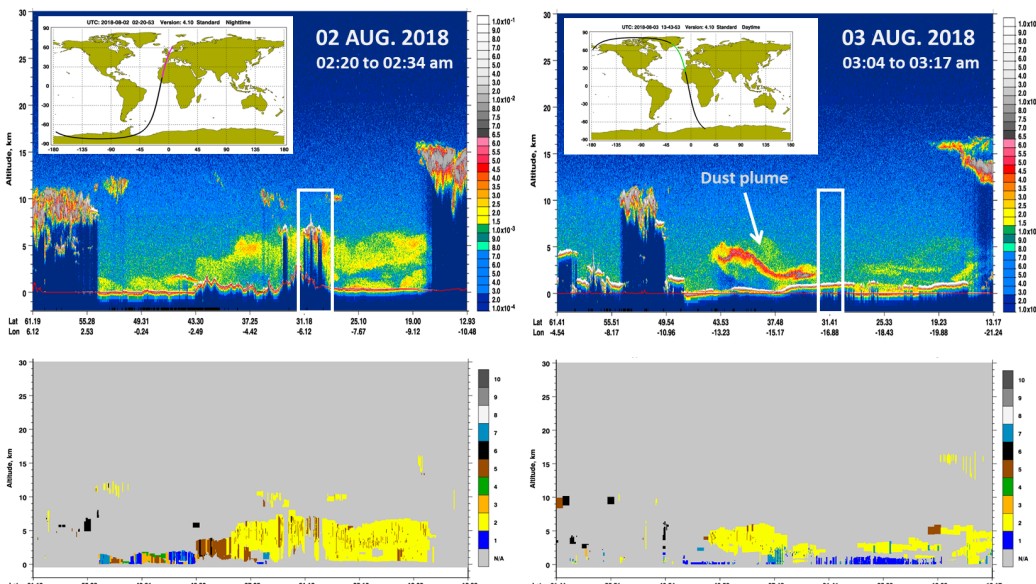

**Figure 7.** CALIOP total attenuated backscatter (km$^{-1}\cdot$sr$^{-1}$) at 532-nm on the upper panels, and aerosol subtype classifications for the same scenes, respectively, observed on 2 and 3 August 2018. The CALIPSO ground track is overlaid in the upper-left corner of panels. The white boxes highlight the nearest CALIPSO flight over the study site. The colours in the subtype images are numbered from 1 to 7 and depict the observed aerosol type: 1 (blue) = clear marine, 2 (yellow) = dust, 3 (orange) = polluted continent, 4 (green) = clean continent, 5 (brown) = polluted dust, 6 (black) = smoke, 7 (light blue) = other.

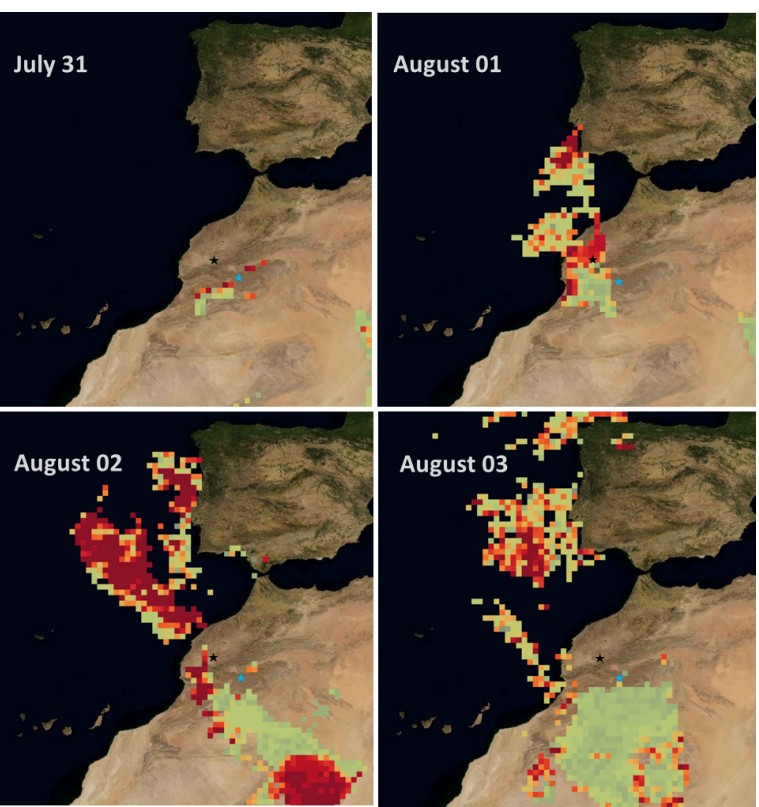

**Figure 8.** Dust Score layers acquired by AIRS instrument on the Aqua satellite, during the dust-storm event, consecutively on 31 July, 1, 2, and 3 August 2018. Pixels where the dust score is less than 400 are not shown. Images are adapted from the NASA Earth Data website (https://earthdata.nasa.gov (accessed on 5 April 2022)). The Saada and Ouarzazate sites are indicated with black and blue stars.

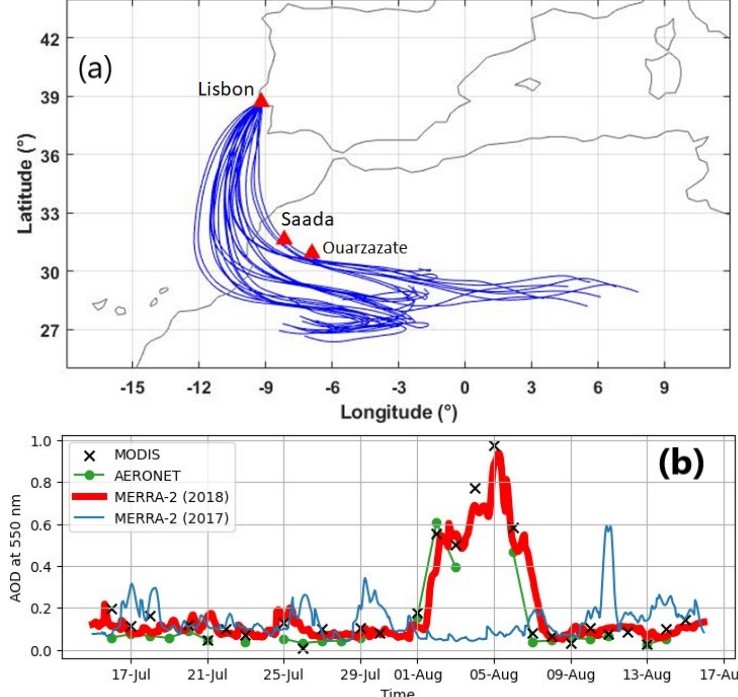

**Figure 9.** (**a**) Ensemble backward trajectories as simulated for 3 days (72 h) by the HYSPLIT model on 2 August 2018. The particles are initially released over Lisbon (38.7°N, 9.2°W) at 4 km above ground level. The red triangles indicate the initialization site (Lisbon) and the observation sites in Morocco,

Saada, and Ouarzazate. (**b**) Daily averaged aerosol optical depth at 550-nm ($AOD_{550}$) from sun-photometer (AERONET) at Cabo da Roca (38.78°N; 9.50°W, 136m asl), 40 km west of Lisbon, together with MODIS (black cross symbols) and MERRA-2 (thick red line) values, from 15 July to 15 August 2018.The blue continuous line shows MERRA-2 $AOD_{550}$ values obtained during the same period, but in 2017.

## 5. Conclusions

The present paper is based on the use of an original aerosol observational time-series in southern Morocco, at the Saada site, located near Marrakech City, 50 km west of the High Atlas Mountains. It is the longest times-series ever recorded in Morocco, covering the period 2004–2019. In addition to Saada data, aerosol observations at the Ouarzazate site, located on the opposite eastern side of the Atlas Mountains, on the edge of the Sahara Desert, have also been used over a shorter period (2012–2016). The local ground-based columnar aerosol optical depth and its variability were investigated together with satellite observations from the MODIS experiment, as well as with MERRA-2 reanalyses. After a seasonal analysis of the aerosol optical and micro-physical properties (AOD, AE, VSD), we analysed in this study a case of a dust-storm event that occurred and crossed southern Morocco in early August 2018. The paper highlights the importance of conducting continuous ground-based observations of the aerosol properties over southern Morocco, a place crossed by Saharan dust plumes especially during the summer season. These observations allow a better understanding and discrimination of the different types of aerosols that can be observed and their variability, which is most relevant to assess aerosol impacts on humans and the environment. This study also showed the role of atmospheric circulation in the long-range transport of aerosols. For that purpose, in addition to ground-based observations, the trajectory and dispersion of the dust plume was investigated from satellite observations (MODIS, CALIOP and AIRS, see Figures 6–8). Moreover, the HYSPLIT model was used to simulate backward trajectories and derive information on where and when the dust plume was transported. A secondary site, Lisbon, was included to consider the extent of the impact of the 2018 dust-storm event. The HYSPLIT simulations confirmed that the trajectories of the air masses at the 4-km layer has crossed southern Morocco over the Saada site. They showed a westward direction towards the Atlantic Ocean, before it changed northward up to the Portuguese coasts (Figure 9). The HYSPLIT simulations also showed that the dust plume was located, 3 days prior, in the south and east of Morocco, and crossed a large area of the Sahara Desert extending from Libya to Morocco.

**Author Contributions:** Conceptualization, H.B.; data curation, H.B. and T.M.; funding acquisition, H.B.; investigation, H.B. and A.B.; methodology, H.B. and N.B.; project administration, H.B.; resources, A.B.; software, N.B. and T.M.; validation, T.M. and K.L.; original draft, H.B.; review and editing, A.B., N.B., T.M., Z.B., T.P. and F.G. All authors have read and agreed to the published version of the manuscript.

**Funding:** This research received no external funding.

**Data Availability Statement:** Not applicable.

**Acknowledgments:** The authors acknowledge AERONET PIs (Bernard Mougenot, Saïd Khabba, Emilio Cuevas-Agullo, and Taoufik Zaidouni) for maintaining the sun-photometers at Saada and Ouarzazate, the National Aeronautical and Space Administration (NASA) for providing MODIS, MERRA-2, CALIOP, and AIRS aerosol and dust data. The authors also thank the NOAA Air Resource Laboratory (ARL) for the provision of the HYSPLIT transport and dispersion model used in the present study.

**Conflicts of Interest:** The authors declare no conflict of interest.

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
