# Peer review of "Aerosol Distributions and Sahara Dust Transport in Southern Morocco, from Ground-Based and Satellite Observations"

_remotesensing, doi:10.3390/rs14102454_

Round 1

Reviewer 1 Report

This is my review on "Aerosol distributions and Sahara dust transport in southern
Morocco, from ground-based and satellite observations"

In this manuscript, the authors present a study of a strong Sahara dust-storm event
that occurred by early August 2018, in the South of Morocco, by the use of aerosol optical depth
(AOD), Angstrom Exponent (AE) and volume size distributions (VSD) as derived from ground-
based observations by 2 AERONET (AErosol RObotic NETwork) sun-photometers at Saada (31.63°N,
8.16°W) and Ouarzazate (30.93°N, 6.91°W) sites, over the 2004-2019 and 2012-2015 periods, respec-
tively.

The subject is very interesting and important, the manuscript is reasonably well written, 
and can be accepted after minor revision.

For example:

> Page 4., line. 136, 142, etc, "Fig.XX"
> In other pages, the authors use "figure" not "Fig." (e.g., line 135), etc....

Taking into account that the range of the particles under study is respirable, I suggest a more detailed discussion 
of the possible destiny of the coarse particles. HYSPLIT calculator have been setted in back-ward mode 
taking Lisbon as final (initial) destiny, but what about near communities or countries? 
What is the probability of sedimentation of a significant burden of that particles?
(we are talking about particulate matter of 2.5 microns), etc...

Reviewer 2 Report

General:

This manuscript presents long-term AOD values from the Aeronet stations at Saada (2004-2019) and Ouarzazate (2012-2015) South of Morocco. Seasonal analysis of the AOD values was presented showing higher values during summer and lower values during winter due to the seasonal distribution of dust storms. Monthly average volume size distributions of the dust particles were also presented showing a higher contribution of smaller particles during summer.

In the paper, a major dust storm event (August 2018) was also presented as an example, and the volume size distribution was compared to the seasonal average of the corresponding month.

During the summer of 2018, the ground-based measurements were compared with satellite-measured AOD values (MERRA & MODIS). The ground-based AOD values were always higher than the satellite data from which MERRA data showed better agreement than MODIS.

The above results made the paper worth publishing after a major revision the work. However, other results like the Hysplit modeling of the dust plume dispersion and the "Lisboa case study" do not bring scientific contribution to the paper. The Hysplit modeling shows the same fact that the satellite images, namely that the plume passed towards the Portuguese coast, thus it is nonsense to present and discuss in the paper. Instead of this I suggest adding more scientific contributions for example the comparison of the AOD value from a Portuguese Aeronet station with the Saada value to estimate the dispersion of the plume. Also, a ground-based PM10 or PM2.5 concentration would be interesting to see how the transported dust plum affects the surface air quality.

See my specific comments in the following:

Abstract: The first sentence is too long. Rewrite.

Ln8-9: the numbers are main diameters or what. Needed to be specified.

Introduction:

Ln31: Obvious sentence. Avoid it.

Ln59: typo: "be-cause"

Ln61: Avoid "production". Dust is rather generated than produced.

Ln63: "Whatever their size...." - already mentioned.

Ln64: "In fact...." - put into a new paragraph. Generally, there are no paragraphs used in the whole manuscript. Fix it.

Ln73: Typo: "Moroc-can".

Ln74: "radii" - we use particle diameter in aerosol science.

Section 2:

Ln110: Indicate the bands here, and then explain which band is used for what (e.g. 940 nm for water vapor).

Ln115: OK, but here you should detail your data, e.g. how many data points per day, what time of the day the measurements were done, length of one data point, averaging time, etc.

Ln117: Why do you mention here? There is no discussion about sky radiation later.

Ln123: What is "level 2.0". Definition needed.

Ln127: From D&K method or AE data? The whole Methodology part is confusing.

Ln128: "climatological" is not the right term. Use "seasonal".

Ln142: "Fig.??"

Ln147: "grey" rather "shaded". See also at Figure 2.

Ln147: "Fig.??"

Ln162: I think "nm" is better to use for wavelength. See Ln180 too.

Ln181: What is Dust Score? Definition needed.

Section 3: Rather "Resutls"

Ln202: seasonal analysis

Ln204: which location.

Ln212: Already mentioned in Ln 204-205.

Ln213: What do you mean? The sentence is not clear.

Ln220: "weaker" rather "lower"

Ln226: "radius" - rather "diameter"

Ln226: "0.15 um" - Figure 4 shows even lower (0.09 um) for summer that is much lower than the used wavelength range (440-870 nm). Do those small particles have measurable extenction at this wavelength?

Ln235: see Ln128.

Figure 3: Be more clear that AOD is at 440 nm, while AE represents the 440-870 range. Also, it would be needed to describe AE meaning in Section 2.

Section 4:

Ln247: Is that 1 measurement point per day or more measurement points per day? What is the time of the measurements? How is it comparable with satellite measurements (same time of the day)? See the comments at Ln 115.

Ln250-252: Only one measurement point I guess. What time is the satellite passage?

Ln261: see Ln226.

Figure 5: Average of how many points? Indicate the time of the measurements. "climatological".

Ln289: Why Lisbon? There is no measurement data presented from Lisbon, what is the reason for the selection? Chose a place where AOD data are available and compare it with the Saada data.

Figure 7, 8 and 9 are meaningless. See my general comment for this section.

Conclusion:

Ln307: see Ln128.

Ln312: typo: "sea-son".

Reviewer 3 Report

In the paper the authors report on the aerosol characterization in two key sites of Morocco, using columnar aerosol properties obtained from AERONET sun-photometers and focussing on the mineral particles contribution in a region mainly affected by dust storm events. A case study is also presented. Data here reported were supported by models outcomes and satellite data in order to define the main aerosol origin and typology. Despite the paper is of some interest, in my opinion the work should be carefully revised.

I suggest to highlight the novelty of the conducted study that is not clear in the text. Moreover, in the introduction the authors should improve the information related to previous studies, stressing general scientific problems and needs and showing results, referring to more recent papers. The comments are sometime superficial and the analysis should be improved (the scatterplot AOD vs AE could be useful). The authors should better discuss obtained results. Finally, I would suggest to revise the English because some sentences are not clear.

The manuscript needs major revision. Main required corrections are described in the annotated manuscript.

Minor comment:

Figure 2: MERRA-2 green line is too thin.

Figure 7: the x-axis labels are illegible. I suggest to insert two different figures.

Figure 8: stars are too small

Please report  in the same way the data in the text, figures and captions.

Round 2

Reviewer 2 Report

The authors have made the necessary revision and corrections.

Reviewer 3 Report

LINE 127 just a typo:

This study is based on ....(please delete  the sentence "reported in the present paper")